# Interventional Radiology in the Management of Metastases and Bone Tumors

**DOI:** 10.3390/jcm11123265

**Published:** 2022-06-07

**Authors:** Ferruccio Sgalambro, Luigi Zugaro, Federico Bruno, Pierpaolo Palumbo, Nicola Salducca, Carmine Zoccali, Antonio Barile, Carlo Masciocchi, Francesco Arrigoni

**Affiliations:** 1Department of Biotechnological and Applied Clinical Sciences, University of L’Aquila, 67100 L’Aquila, Italy; ferrucciosgalambro@gmail.com (F.S.); abarile63@gmail.com (A.B.); carlo.masciocchi@univaq.it (C.M.); 2San Salvatore Hospital, 67100 L’Aquila, Italy; luigi.zugarodoc@gmail.com (L.Z.); federico.bruno.1988@gmail.com (F.B.); palumbopierpaolo89@gmail.com (P.P.); 3Oncological Orthopaedics Unit, IRCCS Regina Elena National Cancer Institute, 00144 Rome, Italy; nicola.salducca@ifo.gov.it (N.S.); carminezoccali@libero.it (C.Z.)

**Keywords:** interventional radiology, bone metastases, tumour embolization, thermal ablation, magnetic resonance imaging-guided high-intensity focused ultrasound, electrochemotherapy

## Abstract

Interventional Radiology (IR) has experienced an exponential growth in recent years. Technological advances of the last decades have made it possible to use new treatments on a larger scale, with good results in terms of safety and effectiveness. In musculoskeletal field, painful bone metastases are the most common target of IR palliative treatments; however, in selected cases of bone metastases, IR may play a curative role, also in combination with other techniques (surgery, radiation and oncology therapies, etc.). Primary malignant bone tumors are extremely rare compared with secondary bone lesions: osteosarcoma, Ewing sarcoma, and chondrosarcoma are the most common; however, the role of interventional radiology in this fiels is marginal. In this review, the main techniques used in interventional radiology were examined, and advantages and limitations illustrated. Techniques of ablation (Radiofrequency, Microwaves, Cryoablation as also magnetic resonance imaging-guided high-intensity focused ultrasound), embolization, and Cementoplasty will be described. The techniques of ablation work by destruction of pathological tissue by thermal energy (by an increase of temperature up to 90 °C with the exception of the Cryoablation that works by freezing the tissue up to −40 °C). Embolization creates an ischemic necrosis by the occlusion of the arterial vessels that feed the tumor. Finally, cementoplasty has the aim of strengthening bone segment weakened by the growth of pathological tissue through the injection of cement. The results of the treatments performed so far were also assessed and presented focused the attention on the management of bone metastasis.

## 1. Introduction

Interventional radiology deals, in the context of radiodiagnostics, with more invasive diagnostic and therapeutic procedures than the execution of diagnostic examinations, for example with computed tomography (CT) and magnetic resonance imaging (MRI). It involves a surgical-like approach but with a minimally invasive profile because the procedures are usually performed by specific needles and/or through vessels.

IR in malignant bone lesions is mainly aimed at the treatment of bone metastases due to the clear epidemiological prevalence compared to primary tumors.

The incidence of ***metastatic bone disease*** is approximately 280,000 new cases per year [1]. Bone is the third most common site of metastatic disease after lungs and liver. Bone metastasis (BM) is the major cause of morbidity in patients with cancer. Apart from pain, complications like pathological fractures and spinal cord compression may occur, worsening patients’ quality of life and prognosis. Fifty percent of the painful symptomatology experienced by patients with cancer originates from BM [1,2,3,4]. In case of tumoral extension to neural structures, pain may be radicular (exacerbated by percussion or palpation) and/or mechanical (exacerbated by movement) [5,6]. In advanced stages of the disease, pain can become intolerable and refractory to conventional therapies, causing walking disability, psychological and functional impact, and significantly impairing quality of life [2,3,4,7]. The approach to the patient with BM must be strictly multidisciplinary with the involvement of all professional figures (oncologist, radiotherapist, orthopedic, and pain therapy specialist) to offer the patient the best treatments. In the decision making for surgical and/or medical treatment, life expectancy of patients with metastatic bone disease is the most important factor; a short life expectancy suggests the use of less invasive procedures, that bring lower complication rates and shorter rehabilitation times [7]. In this context, interventional radiology procedures play an important primary or complementary role in the management of BMs [8,9,10], especially in terms of palliation of pain and treatment of fractures. Short life expectancies more often require palliative rather than curative regimens. Therefore, rapid pain relief has become a priority. Several conventional treatment options have been described, including opioids, hormone therapy, chemotherapy, radiation therapy, and surgery, all of which have side effects and contraindications. Radiation therapy (RT) remains the gold standard, but up to 20% of patients do not respond; moreover, the greatest reported benefit is received only after 5–20 weeks from completion of treatment [6,11,12]. Over the last two decades, percutaneous image-guided interventional techniques have emerged with satisfactory results in the management of painful bone metastases. The thermal ablation techniques allow loco-regional tumor ablation while cementoplasty stabilizes the bone segments weakened by pathological tissue [13,14,15]. These techniques can also be used with a combined approach [16,17,18]. Percutaneous bone ablation is a minimally invasive treatment offering several advantages. It is repeatable, does not need skin exposure, as in the case of radiotherapy and has no interference with systemic treatments.

Aim of our study was to evaluate, through the review of the recent literature, the efficacy and duration of pain and local tumor control using interventional radiology procedures in the treatment of both primary and metastatic bone tumors.

***Primary bone tumors*** are relatively rare compared with secondary (skeletal metastatic) disease [19,20,21]. They represent less than 1% of all cancers in adults. Usually, IR in this field is reserved to non operable patients or adiuvant treatments.

## 2. Interventional Radiology—Techniques

The most common techniques for management of patients with metastatic bone disease include embolization and thermal ablation therapies (Table 1).

**1.** 
**Embolization**


Embolization is performed for the treatment of bone lesions either alone or in combination with other techniques [2,3,7]. It reduces hypervascularization of the lesions by injection of embolic agents into the vessels, causing necrosis of the tumors. The embolic agents can be used for preoperative transarterial embolization (TAE) and for palliative cases. Nano-particles are most used agents for bone devascularization. The choice of particle diameter (40–1200 micron) depends on vessel size and desired distal embolization [5,22]. This procedure is performed in the angio-suite. A peripheral artery is cannuled (most commonly femoral or radial artery). Navigating through the arterial circulation, the lesion is reached and the arterial vessels that feeds the lesion are embolized.

Indications

TAE can be used in case of hyper-vascular BM for the following scenarios:To reduce blood deficit during surgery; in this case, TAE should be executed within 3 days from surgery to lower the risk of tumor revascularization [3,4].To lower pain and natural bleeding of tumors that cannot gain benefit from surgical or percutaneous therapy.To reduce tumor vascularization before percutaneous ablation and reduce the heat/cold-sink effect.

**2.** 
**Thermal ablation**


Percutaneous thermal ablation was introduced in the clinical practice as a palliative treatment of painful bone metastases in the early 2000 [23,24]. It is defined “percutaneous” because the energy pass through the skin, the soft tissues and the bone using a needle or a needleless technique (magnetic resonance imaging-guided high-intensity focused ultrasound MRgFUS). The most used techniques include radiofrequency thermal ablation (RFA), cryoablation (CA), laser and microwaves thermal ablation (MWA) [7]. The main advantage is the possibility to create an ablation area of known and reproducible size. Ethanol injection is a precursor of thermal ablation and is considered a technique of chemical ablation: through direct injection of alcohol into the lesion, it produces tumor necrosis directly through cellular dehydration and indirectly through vascular thrombosis and tissue ischaemia. This technique does not allow full control of the ablation, since the diffusion of ethanol is not fully predictable or reproducible [7].

Thermal ablation causes a fast change in the temperature of cells, thus damaging their membranes and generating necrosis or coagulation or both. We can distinguish two types of techniques: “needle driven”, when a needle is inserted into the lesion to drive energy (RFA, MW, or CA) or needleless technique, like High Intensity Focused Ultrasound (HIFU), where an ultrasound beam passes through the tissue and is focused on the target causing the ablation; in this case no needles are used.

### 2.1. Radiofrequency Ablation

RFA is the most widely employed thermoablation method with proved high success rates, not only in the treatment of liver, lung, and kidney tumors, but also for the treatment of bone and soft tissue tumors, for skeletal issues and local tumor management [23]. RFA is usually employed in combination with CT for a rapid and correct placement and control of the ablation electrodes [9]. The procedure is painful, therefore patients are usually under general anesthesia or periferical blocks. RF probes with active tips able to create different sizes of ablation (from 0.5 to 4–5 cm) are typically positioned in the center of the tumor. If some critical structures (like nerves or spinal cord in case of vertebral lesions) are too close to the lesion, the area of ablation can be reduced. This may increase the risk of relapse or uncomplete treatment [6,10]. Before ablation, it is possible to obtain biopsy samples using a coaxial approach. Ablations are performed for a total of 6–10 min at a temperature of 80-to-95 °C, depending on the manufacturers. RFA can also be safely combined with cement injection (cementoplasty), with excellent results in pain palliation and bone stabilization at various skeletal sites, in particular the vertebral bodies [13,25,26,27,28,29]. When treating bone metastasis, the role of RFA is not only limited to pain palliation; the technique is in fact offering promising results also in local tumor management. In this case, the ablation area should exceed the whole lesion of some millimeters [13,27,30]. This last rule should be observed also in case of treating bone tumors others than bone metastasis. On the other hand, in case of benign lesions (osteoid osteoma OO, osteoblastoma OB, etc.) only the area covered by the lesion is treated.

Indications:Benign tumors (OO, OB, desmoid tumors, chondroblastoma, and gigantocellular tumors of the bones GCTs [31,32,33,34]).Malignant tumors (skeletal metastases, myeloma, soft tissue metastases, and plasmocytomas) also combined with cementoplasty [13,35,36,37].Neurovascular diseases related to the musculoskeletal system: Use for palliative pain relief; targeting locoregional neural supply, neuromas, neuroendocrine metastases, hemangioendotheliomas, and intramuscular hemangioma [38,39].

### 2.2. Microwave Ablation (MWA)

MWA can induce effective coagulation quicker than other ablation techniques and spread out deeper thanks to the characteristics of the created energy. It is less time consuming and provides better results on deeper and larger lesions than other ablation techniques [40,41,42]. As in RFA, an ablation needle is introduced into the center of tumors and MWA is performed with variable energies and times (averagely, from 3 to 10 min) depending on the size of the lesion. Compared to other ablation techniques, the penetration of MWA is deeper, and more resistant to the effects of heat buildup and charring [43,44,45].

Indications:Benign OO: MWA can reliably treat OO, with no recognized complications or recurrence; less evidence then RFA.Malignant (skeletal metastases, multiple myeloma, soft tissue metastases, and plasmocytomas).

### 2.3. Cooling Techniques: Cryoablation

Cryoablation freezes the tumour core with extremely cold temperatures. The tip of the probe employed generates temperatures that reach −40 °C and even less. A −20 °C temperature induces cell destruction and trigger apoptosis and focal sudden cellular ischemia. The gas used to trigger freezing is argon or helium (both inert gases) delivered through small cryoprobes to induce rapid freezing and thawing of target tissues according to the Joule–Thompson effect [46,47,48]. CT and MRI are preferred as techniques of guidance because it is possible to monitor the area of ablation that is called “ice ball” to ensure a full coverage of the tumor and minimizing complications to the surrounding vital structures [49,50].

Indications:Benign: Extra-abdominal desmoid tumors, OO, OB, Aneurismal Bone Cists, primary bone tumors, neuroma and arteriovenous malformations (AVMs) and chondroblastoma.Malignant: Skeletal metastases (even osteoblastic and sclerotic; myeloma, soft tissue metastases, plasmocytomas, and in-transit melanoma metastases.Neurovascular: Palliative; neuroendocrine metastases, hemangioendotheliomas, and benign neural tumors such as Morton’s neuroma.

### 2.4. Non-Percutaneous Thermal Ablation Procedures: High-Intensity Focused Ultrasound Ablation (HIFU)

HIFU uses high intensity focused ultrasound beams, that act as vehicle of energy and burn the lesion in depth. Combined with MRI, the technique is called Magnetic Resonance Imaging-Guided High-Intensity Focused Ultrasound (MRgFUS) [43,51]. MRgFUS is a totally non-invasive method of ablation able to reach temperatures up to 90 °C, resulting in coagulative necrosis [52,53]. So far, this technique has been applied in the treatment of diseases involving bladder, prostate, uterine tumors, and bone metastases. MRgFUS is a safe and effective non-invasive treatment option for pain-inducing metastatic bone lesions refractory to radiation therapy with >70% of patients experiencing variable pain reduction [54,55]. Some limits are represented by presence of thick cortical bones (being an obstacle to the propagation of the ultrasound beam), patients with great mass index increasing the distance between the skin surface and the lesion, and lesions situated <1 cm from the skin increasing the risk of skin burns. In real time during the procedure, it is possible to measure the temperatures reached in the focal zone of ablation and verify whether more energy is needed to perform the procedure successfully [56,57,58].

Indications:Benign: OO and OB can be successfully treated with MRgFUS with complete pain relief and no morbidity; extra-abdominal desmoid tumors.Malignant: metastases and/or multiple myeloma, plasmocytoma, and other focal myeloproliferative disorders

### 2.5. Stabilizers

#### 2.5.1. Cementoplasty

The procedure consists of the injection of polymethyl methacrylate (PMMA) within a vertebra or another bone segment fractured or weakend by pathological tissue. The cement strengthens the bone and reduces mechanic pain. PMMA is injected once the needle is positioned in the bone lesion (using fluoroscopy or CT). The consistency of the cement increases slighthly and hardens after approximately 10 min during the polymerization phase. This is accompanied by an exothermic reaction with the temperature peak increasing up to 75 °C in the middle of the treated bone [59,60]. Vertebroplasty is currently the most adequate local treatment to quickly stabilize vertebral fractures or vertebrae at risk of fracture, to reduce pain and improve quality of life [61,62]. Cementoplasty is contraindicated in patients with coagulation problems, extensive osteolytic destruction, especially on the posterior border of the target vertebral body, poor general conditions, periodic presence of local or systemic infections, allergy to bone cement, and asymptomatic vertebral compression [63,64,65,66]. Because vertebroplasty is only aimed at treating the pain and consolidating the weight-bearing bone, other specific tumor treatments should be performed in combination for tumor management [13,67,68,69], in particular ablation for bone tumors.

The major indications for vertebroplasty in oncological settings are:Patients with severe pain and neurological damage caused by spinal lesions, and resistance to conventional treatments.Patients with spinal instability caused by spinal lesions within the vertebral body.Patients with contraindications to open surgery.

#### 2.5.2. Percutaneous Osteosynthesis

Percutaneous osteosynthesis consists in the insertion of screws for the fixation of minimally displaced or non-displaced fractures, and consolidation of fractures [70,71]. The indication is reserved to cancer patients, who are not candidates for surgery and have limited life expectancy [72]. The procedure is performed under fluoroscopic guidance or CT, with planning of the screw trajectory and skin entry points. The procedure takes approximately two hours, therefore general anaesthesia is usually performed. Surgery is preferred to percutaneous osteosynthesis whenever possible since the long-term effectiveness of the latter is still to be investigated [73,74].

### 2.6. General Requirements Needing Careful Analysis before IR Procedures

Despite the goal of treatment, metastatic bone disease does not require biopsy before interventional sessions, unless:-The primary cancer is unknown;-More than two primary tumors are suspected;-Specimen analysis is crucial to adjust systemic therapy.

In these cases, a biopsy can be easily performed prior to each percutaneous procedure [75].

Before undergoing IR procedures, each patient should have normal recent laboratory tests including blood cells count, coagulation tests (prothrombin time, activated partial prothrombin time and international normalized ratio) and normal kidney function. In case of known or suspected local or systemic infection, any type of IR procedure should be postponed until the infection is over. Each IR bone procedure needs a strictly sterile environment and prophylactic intra-operative antibiotic coverage (1 g cefazolina ev).

### 2.7. Follow-Up after IR Procedures

Clinical and radiological follow-up should be scheduled on a regular basis according to the aim and type of treatment. It is important that radiological follow-up be performed by dedicated radiologists having at least basic knowledge in the field of extra-vascular IR.

Despite the goal of treatment, clinical evaluation should assess any change in pain experienced by the patient as compared to the pre-treatment status. The most applied test is the 0–10 points VAS. Furthermore, patient independence in daily life activities and quality of life should be evaluated by applying dedicated tests like the Functional Independence Measure or the Karnofsky Performance Score. Additionally, some authors also evaluate the consumption of analgesics including opioids [12,76] limited by side effects such as constipation, sedation, and nausea.

In case of palliative treatments, clinical evaluation may be sufficient (unless unexpected adverse events are suspected) and clinical and radiological follow-up should be adapted according to the specific local evolution of the treated lesion. For instance, clinical suspicion of secondary fracture needs careful investigation with imaging exams (X-ray or CT). No contrast medium administration is needed if CT scan is applied.

On the other hand, radiological assessment is mandatory to evaluate the results of curative treatment by depicting any residual viable tumor. In this perspective, contrast- enhanced imaging (CT, MR, or PET-CT) is needed to exclude any tumor size increase and contrast medium uptake, as compared to baseline imaging. The techniques of choice are MR and PET-CT.

## 3. Literature Search Strategy and Results—Interventional Strategies in Bone Metastases

Research was conducted on PubMed library with the following key words “Interventional radiology of primary bone tumour” and “Interventional radiology of bone metastases”. Filters applied were publication years between 1990 and 2022 and other languages except English. Duplicate results were excluded. Abstracts were excluded. In the initial screening, 3508 articles were identified. We excluded 3376 articles because the treatments were not performed percutaneously. We also excluded meta-analysis and systematic reviews. The resulting 132 articles were divided into three groups: curative treatment (*n* = 15), palliative (*n* = 111), and primary bone tumor (*n* = 6). A flowchart summarizing study selection is shown in Figure 1. Baseline characteristics of included studies are summarized in Table 2, Table 3, Table 4 and Table 5. The analysis was performed following the Quality Improvement guidelines for bone metastasis management issued by the Cardiovascular and Interventional Radiological Society of Europe (CIRSE), indicating that interventional treatments not only offer local tumor control, but also effective pain palliation [75]. All the results of our research are listed in the Table 2, Table 3 and Table 4; the most recent results are discussed in each section.

### 3.1. Curative Treatment

Indications

According to guidelines of the Cardiovascular and Interventional Radiological Society of Europe (CIRSE), the curative treatment of bone metastases is defined as complete and definitive ablation of the tumor [75]. Candidate patients for these treatments are oligometastatic patients (<3 metastases, each <3 cm in size) with favorable prognostic factors such as young age, absence or limited destruction of cortical bone, absence of extraosseous metastases, good performance status and slow evolution of the underlying disease. Among oncological patients, few patients have these characteristics. Curative treatment can be achieved with percutaneous thermal ablation performed by means of RFA, MWA, or CA. To achieve effective local control, the margin of safety of tumor ablation must be at least 0.5–1 cm larger than the lesion. Combined treatment may be used, including embolization, but a technique of ablation should always be used to ensure the complete destruction of the tissue treated. After each treatment follows the assessment of local progression-free survival (LPFS), i.e., the time interval between locoregional treatment and the development of local tumor progression at the treated site; bone disease-free survival (BDFS—defined as the time interval between locoregional treatment and disease progression in the treated BM or another bone); disease-free survival (DFS—defined as the time interval between locoregional treatment and the time of any visceral or bone tumor progression). Finally, overall survival (OS).

Available evidence

**Arrigoni et al.** [87] retrospectively analyzed the results of 28 cryoablation procedures of BM. Of them, 11 patients were treated for local tumor control. In a mean follow-up of 22.4 months, it was recorded stability and/or reduction of lesion volume in 10 of 11 patients.

**Cazzato et al.** [77] reviewed from February 2009 to June 2020, 23 patients with sacral metastases undergoing RFA or CA with palliative or curative intent. In patients undergoing curative ablation (*n* = 7), local tumor progression occurred at 3 (43%) treated sites at a median follow-up of 17 months.

**Ma et al.** [78] performed a retrospective review (January 2011 to April 2016) of 76 BM in 45 patients treated with percutaneous ablation. A total of 48 out of 76 BM (63%) were treated with RFA, 35% (27/76) with CA, and 1.3% (1/76) with MWA. In 70% (53/76) of cases, an associated cementoplasty was performed. After 3 months, local tumor recurrence was documented in seven cases. No tumor progression in the treated area was documented at 6-month follow-up, and after 1 year, local tumor control for 17 tumors was 68% (17/25). There were no procedure-related complications for metastases treated with RFA, according to the Society of Interventional Radiology (SIR) guidelines. For metastases treated with CA, the complication rate was 7.4% (2/27).

A retrospective analysis by **Vaswani et al.** [79] of 64 bone metastases from sarcoma in 41 patients treated with ablation between December 2011 and August 2016 was performed. Two subgroups were treated: oligometastatic disease (*n* = 13) and extensively metastatic disease (*n* = 51). One hundred percent (10/10) of the treated lesions showed local tumor control in oligometastatic disease at 1 year. Three of thirteen ablated lesions were lost at follow-up at 3, 6, and 9 months, respectively.

**Cazzato et al. [28]** retrospectively reviewed 46 patients treated for BM from January 2008 and November 2017. Complications were observed in 20.4% of cases (*n* = 10;): out of whom, only two required interventional or surgical treatments; in the other cases, only postoperative pain (grade 2 complication) was observed. The mean follow-up was 34.1 ± 22 months (median 30.5; 95% confidence interval [CI], 27.6–40.7). Local tumor progression at the treated site was noted in 28.5% of cases (*n* = 14); 1- and 2-year survival was 76.8% and 71.7%, respectively. Only BM size >2 cm was associated with local tumor progression. BDFS was 71.7% and 53.1% at 1- and 2-year follow-up, respectively. At the same time interval, DFS was 86.3% and 61.5%, respectively. Finally, 1- and 2-year OS was 95.4%.

**Gardner et al.** [15] retrospectively reviewed 40 patients with metastatic renal cell carcinoma who underwent cryoablation for BM between 2007 and 2014. Complications included five delayed fractures in four patients who were seen 1 month to 8 months after cryoablation. Only one of these delayed fractures was symptomatic and required treatment with cementoplasty for pain control. The overall local tumor control rate for metastases was 82% (41 of 50). The likelihood of local recurrence was 25-fold greater for patients with disseminated disease than for patients with oligometastatic disease (*p* = 0.001). Despite a local tumor control rate of 82%, 38 of the 40 patients had evidence of new metastases or disease progression to other sites at the end of the follow-up period, and the median time to progression was 6 months (95% CI, 4.0 to 11.0 months).

**Eire et al.** [80] retrospectively analyzed 16 patients with oligometastatic prostate cancer who underwent image-guided percutaneous ablation to treat 18 metastatic sites between 2009 and 2014. All but one were bone lesions; in the study a pelvic lymph node was included. Local tumor control was achieved in 15 out of 18 metastases (83%) at a median follow-up of 27 months after ablation. Local tumor recurrence occurred in three out of 18 metastases (17%). PFS rates at 12 and 24 months were 56% (95% CI, 30–76%) and 43% (95% CI, 19–65%), respectively. Among the 18 image-guided percutaneous ablations, no complications of grade 3 or higher according to the Common Terminology Criteria for Adverse Events were observed.

**Wallace et al.** [13] retrospectively reviewed 55 spinal bone metastases undergoing RFA and vertebral augmentation of bone metastases between April 2012 and July 2014. Five cases of residual tumor were documented within 3 months of treatment. In all five of these cases, there was also evidence of systemic disease progression. The radiographic control of local tumor at 3 months was 89%. The rate of radiographic tumor control 6 months after treatment was 74% (26/35) and after 1 year was 70% (21/30) and 67% (18/27) in cases of systemic progression of metastatic disease.

**Deschamps et al.** [83] retrospectively analyzed 122 patients who underwent thermal ablation of BM between September 2001 and February 2012. In multivariate analysis, factors associated with a lower risk of treatment failure were metachronous bone metastases (*p* = 0.004), no cortical bone erosion (*p* = 0.01), maximum diameter at baseline CT < 20 mm (*p* = 0.001), no nearby critical neurological structures (*p* = 0.002). The 1-year OS rates were 91% (95% CI: 80–97%) and 90% (95% CI: 75–97%) for patients in group 1 (in patients with oligometastatic disease) and group 2 (in patients with long life expectancy), respectively. For patients in group 1, the B-DFS was 64% (95% CI: 50–76%) 1 year after thermal ablation of all bone metastases. Major side effects occurred in 4 patients (3%) of the 122 thermal ablations performed: avascular necrosis of the femoral head, nerve root injury (2 patients), and stress cardiomyopathy (Tako–Tsubo) immediately after thermal ablation of a bone metastasis from a pheochromocytoma. Seven patients (8%) experienced SREs despite thermal ablation. Most of the SREs were fractures at the treated site. One patient experienced spinal cord compression that occurred 8 months after treatment failure of a bone metastasis located at T5.

### 3.2. Palliative Treatments

Indications

Palliative treatments can be offered to most patients with painful bone metastases (at least 6–7 on a VAS 0–10 scale). The goal is not complete ablation of the tumor but pain palliation, tumor size reduction (debulking), prevention of pathological fractures and/or decompression and debulking of tumors (especially for spine tumors protruding into the spinal canal). If pain management is necessary and there is no risk of fracture, only thermal ablation should be performed (such as RFA, MWA, CA, or MRgFUS). A combined percutaneous cementoplasty procedure is performed in case of fracture or impending fracture. Tumors with less than 1 cm from the critical structures like spinal cord or large blood vessels are a relative contraindication if these structures cannot be isolated and protected (with air, CO_2_, or glucose).

Available evidence

**Arrigoni et al.** [87] retrospectively evaluated 17 patients with single bone lesion treated by CA with a palliative intent. In a 3-month follow-up study, we recorded an overall reduction of pain (evaluated using a VAS 0–10 scale) between pre- and post-treatment from 7.4 to 4.5. No major complications were recorded.

From February 2016 to March 2018, **Jenning et al.** [88] performed a multicenter, prospective, single-arm study of 64 patients with painful metastatic bone disease, who were not candidates for or had not benefited from standard therapy. In an analysis of the 64 patients treated with CA, for whom complete follow-up data were available, response rates of pain over time ranged from 38% to 48% over the 24-week follow-up period from week 1 to 24 using the Brief Pain Inventory-Short Form (reference range 0 to 10 points). Response rates were similar among participants who had and had not undergone previous radiation per week. Opioid medication use decreased significantly in patients from week 4 to week 24, as did quality of life, which improved steadily over the first 6 months. Possibly related adverse events occurred in 22% (14 of 65) of cases. Three of the 65 participants (4.6%) each had a grade 3 or 4 serious adverse events (abdominal pain, hematoma, and skin burn or frostbite).

**Levy et al**. [14] analyzed one hundred and six patients between October 2017 and March 2019. After RFA, the patients experienced significant improvement in worst pain, average pain, pain interference, and quality of life. Before RFA, the mean worst pain score was 8.2. After ablation, worst pain improved significantly, with mean scores falling to 3.5 at 6 months (*p* < 0.0001). More than half of patients (59%) experienced immediate improvement, reporting a 2-point change in worst pain at targeted treatment sites 3 days after ablation. The mean 24-h morphine equivalent dose for all treated subjects at baseline was 61.0 mg and decreased to 50.4 mg at the 3-month visit.

**Yang et al.** [92] observed a total of 26 patients (with 36 metastatic bone lesions) who were treated with CA between May 2012 and June 2016. The overall pain response rate was 91.7%, 94.4%, 91.7%, and 94.4%, respectively, at 1 day, 1 month, 3 months, and 6 months after CA, with CR achieved at 22.2%, 41.7%, 36.1%, and 22.2%, respectively. Pain relapse occurred in four patients, one of whom had a pathologic fracture. Two of these patients were treated again with CA. Adverse events were observed in 3 out of 26 patients.

Thirty-three patients with metastatic bone tumors were enrolled by **Tanigawa et al.** [96]. No serious complications occurred during the study. Four patients showed adverse events, including one case each of grade 3 pain and grade 2 hypotension, and two grade 1 burns. The response rate was 69.7% (95% confidence interval: 51.3–84.4%). The dose of analgesics was reduced within 4 weeks in 13 patients but had to be increased in 2 patients. Clinical efficacy was assessed by intention-to-treat analysis and was 69.7%. Although this efficacy rate is lower than that reported in previous retrospective studies [7,20,21], it is acceptable for the present study because it was a prospective investigation. In addition, the changes in VAS demonstrate that the analgesic effects were both immediate and long-lasting.

**Prologo et al.** [97] retrospectively analyzed percutaneous CT-guided CA in the setting of painful musculoskeletal metastatic disease in 50 patients. There were statistically significant decreases in median VAS and narcotic use at both 24 h and 3 months. The median VAS score of all patients reported before surgery was 8 ± 1. At 24 h and 3 months after the procedure, the median VAS score decreased to 3 ± 1 (*p* < 0.000). Morphine equivalent values decreased to 85 ± 70 mg (*p* < 0.000) 24 h after the procedure and remained significantly lower than before the procedure (*p* < 0.000) after 3 months. In six patients (11%) two minor complications were observed, and four major consisting of two complete femoral fractures after ablation and cementoplasty of proximal lytic lesions, and two transient neuropathies resulting from damage to adjacent nerves during ablation.

From July 2011 to January 2012, **Pusceddu et al.** [76] analyzed 18 patients with symptomatic skeletal metastatic lesions undergoing treatment with MWA. Seven patients had been previously treated with RT, three patients with RT and chemotherapy, and seven with chemotherapy alone. In all these patients, pain had proven refractory to conventional approaches. Technical success was achieved in 100% of the cases. Postprocedural CT scans demonstrated no major complications. The mean BPI (Brief Pain Inventory) score, 7 days after the procedure, was reduced by 77%. Four weeks after treatment, nine out of 18 patients (50%) were completely pain free, and the remaining patients reported a 64% reduction in BPI score. Twelve weeks after the MWA procedure, the mean BPI score was reduced by 92% (range, 41–100%). Thirteen out of 18 patients (72%) were symptom-free and did not resume any therapy.

Regarding embolization as a palliative and/or adjuvant treatment of bone metastases the literature is smaller compared with ablative procedures. One of the larger series of cases was collected by **Rossi et al.** [98] that retrospectively studied 107 patients with BM from renal cell carcinoma treated from December 2002 to January 2011 with 163 embolizations with N-2-butyl cyanoacrylate (NBCA). The mean tumor diameter before embolization was 8.8 cm. A clinical response (defined by a reduction of at least 50% in pain and a reduction of at least 50% reduction in analgesic doses) was achieved in 157 embolizations (96%) and no response in six embolizations of sacroiliac metastases. The median duration of clinical response was 10 (range 1–12) months. Areas of hypoattenuating signal on CT (tumor necrosis index) were observed in all patients. Variable ossification was observed in 41 patients. The mean maximum tumor diameter after embolization was 4.0 cm.

### 3.3. Palliative Treatment of Both Bone Fracture and Impending Fracture

Indications

Patients with BM and impending pathologic fractures may be selected for possible treatment with cementoplasty.

Osteolytic tumors (e.g., metastases, lymphoma, multiple myeloma) located in the vertebrae, acetabulum, or condyles, causing local pain, disability, or being at risk for compression fracture are the primary indications for cementoplasty.

It is critical to determine whether compressive or torsional forces are predominant in the involved bone.

In cases where compressive forces are involved (i.e., spine, acetabulum, femoral condyles, tibial extremities, talus, and calcaneus), percutaneous injection of Poly Methyl MethAcrylate (PMMA) cement allows for bone consolidation and prevention of pathologic fractures. Due to its properties, PMMA leads to pain relief. PMMA polymerization is an exothermic process with a transient increase in temperature. This exothermic reaction has a direct toxic effect on the nociceptors; furthermore, the cement stabilizes the intratumoral microfractures, which are sources of pain in metastatic fractures. PMMA is injected as a dental paste material. Once deposited into the bone defect, it solidifies within 20–30 min. The distribution of PMMA within the bone defect is unpredictable [183,184]. However, percutaneous cementoplasty alone does not provide any radical cytotossic effect. Therefore, percutaneous cementoplasty should always be preceded by ablative treatment if a cytotossic effect (palliative or curative) is the aim of the procedure.

In the bony districts where torsional forces are most likely to act (the diaphysis of long bones), percutaneous cementoplasty can be sufficient to achieve pain palliation; however, it does not guarantee a definitive bone consolidation. Therefore, other forms of consolidation (intramedullary nailing or external fixation performed percutaneously or surgically) may be necessary along with cementoplasty. Even for long bone injuries, percutaneous cementoplasty alone (with or without other percutaneous or surgical consolidation strategies) has no curative effect.

Complications reported following cementoplasty are cement leakage from vertebrae, reported in 38% to 72.5% of cases [185]; pulmonary embolism, infection, and fracture in less than 1% of cases; allergic reactions, bleeding from the puncture site, pain in other areas, described in 14% of patients [186].

Available evidence

**Wang****et al.** [16] retrospectively analyzed 35 patients with neoplastic vertebral lesions who received RFA combined with vertebroplasty (group A, 15 patients with 17 lesions) or single vertebroplasty (group B, 20 patients with 24 lesions) from March 2016 to June 2019.

VAS scores in group A decreased more rapidly 1 week after treatments and remained more stable at 6 months than in group B (*p* < 0.05).

Twenty-five patients with refractory painful vertebral metastases were included in the study by **Kastler et al.** [90] consecutively between 2012 and 2019. The decision to perform vertebroplasty was made based on the location, type, and extent of the lesion, and the Kostuik score was used to predict fracture risk.

The mean VAS before the procedure was 8.4/10. Significant pain reduction was achieved in 24/29 (83%) procedures at 1 month.

Therapeutic failures on pain palliation were observed in three procedures with advanced stage of disease. In each case, the lesions were large with significant prevertebral invasion of soft tissues and foramina. In five other patients, a decrease in pain of less than 50% was observed: at 1 month (one patient), at 3 months (two patients), at 6 months (two patients), and at 1 year (two patients). Significant growth of tumor lesions or new lesions was observed in these patients.

Approximately 82% of patients had a satisfactory analgesic outcome at 1 month with significant long-term pain palliation. Pain relief was obtained immediately after the procedure and provided significantly lasting pain relief. Therefore, end-of-life quality was improved in these patients suffering from intractable pain.

**Madani et al. [89]** between June 2016 and January 2021 retrospectively examined treatment complications, pain palliation, and skeletal complications after combined local treatments (CLT) for vertebral metastases with limited epidural extension (VMLEE).

Eighteen consecutive patients underwent CLT for 24 VMLEE, between June 2016 and January 2021. No major post-treatment complications were reported. Only one vertebra showed an increase in a preexisting vertebral fracture. Nine VMLEE had evidence of residual disease, including two that resulted in spinal cord compression (2, 11 months).

**Pusceddu et al.** [91] examined 35 patients with 41 spinal vertebral metastases undergoing targeted percutaneous radiofrequency ablation with a navigable radiofrequency ablation device associated with vertebral augmentation. Twenty-one patients (60%) had one or two metastatic lesions (Group A) and fourteen (40%) patients had multiple (>2) vertebral lesions (Group B).

The procedure was technically successful in all treated vertebrae. The VAS decrease over time between 1 week and 1 year after radiofrequency ablation was similar, suggesting that pain relief was immediate and durable.

From 1 October 2017, to 1 October 2019, **Jiao et al.** [18] analyzed 30 patients with 42 metastatic osteolytic or mixed osteolytic tumors, treated with percutaneous microwave ablation and percutaneous cementoplasty simultaneously under CT and scopic guidance to control local lesions and severe pain.

Opioids were discontinued in nine cases (30%) at 12 weeks. In five cases (16.7%), opioids were replaced with nonsteroidal anti-inflammatory drugs. QoL assessments at 4 weeks showed significant improvements in physical function, body pain, general health. Vitality (*p* < 0.01), social function (*p* = 0.002), role emotion (*p* = 0.02), and mental health (*p* = 0.007) showed significance at 12 weeks compared with pre-treatment values.

None of the patients showed serious complications. Asymptomatic bone cement loss occurred in two cases with extraspinal metastases and in one case with spinal metastases.

**Yang et al.** [93] conducted a retrospective study of 63 patients with osteolytic vertebral fractures treated by vesselplasty (a type of minimally invasive stabilization system for vertebral fractures) from September 2014 to January 2018. The rate of pain relief considered as VAS and ODI scores was 97.4% in 37 out of 38 patients at 1 year after surgery. The rate of bone cement loss was 16.2%, of which 3.8% (4/105) of the segments leaked into the spinal canal but without nerve symptoms. No cases of infection occurred.

A total of 274 patients with 367 bone metastases undergoing percutaneous RFA and CA between January 2008 and April 2018 were reviewed by **De Marini et al.** [94]. Metastases treated with RFA were more frequently treated in combination with cementoplasty during the same session 20/66 (30.3%) versus 64/301 (21.3%) of cases treated with CA (*p* < 0.001). Complications occurred in 49/367 (13.4%) of the bone metastases (major, 2.5% of cases, minor complications in 10.9%). Complications were more likely with RFA than with CA (23/66 [34.8%] versus 26/301 [8.6%], respectively; *p* < 0.001); minor complications (immediate post-procedural pain) occurred more frequently with RFA (22/66 (33.3%)) compared with CA (18/301 (6.0%); *p* < 0.001). Major complication rates were similar between RFA and CA (RFA 1/66 (1.5%) vs. CA 8/301 (2.7%); *p* = 1). RFA and CA were comparable for treatment efficacy (curative/palliative ratio 14/52 (26.9%) vs. 47/254 (18.5%), respectively; *p* = 0.27.

**Deib et al.** [95] performed a retrospective review that included 65 patients with 77 tumors who underwent microwave ablation and cementoplasty for 18 months. Complete ablation was achieved in all 77 tumors.

A mean decrease of 17 (SD, 12.9) ODI points was observed in the interval between before the procedure and 20–24 weeks after the procedure. A mean decrease of 4.30 (SD, 3.32) VAS points was observed between before and 20–24 weeks after the procedure.

### 3.4. Role of Radiation Therapy and Surgery

The gold standard treatment for painful BM is radiation therapy (RT) [187,188], a well-accepted local treatment modality. The aim of palliative radiation therapy is pain control, prevention of pathological fractures, and spinal cord compression to improve quality of life, reduce the need of analgesics, and, in some cases, improve the survival rate. In presence of no response to RT, partial response, or recurrence of pain, re-irradiation of the same area may be considered [189].

Re-irradiation is usually required in a substantial group, considering that 40% of patients achieve no pain relief after initial RT. Moreover, pain comes back in approximately 50% of patients within one year of RT.

Ablative approaches have been proposed as a therapeutic option to control painful BM [127,190], even in combination with RT. Life expectancy is crucial in the choice of the best surgical treatment of BM. Short life expectancy, due to histology, staging and conditions of the patient, requires palliative treatments. Good life expectancy may require more aggressive treatments of metastases, performed to last over time (excisional surgery) with radical intent [191,192].

Pathological fractures or injuries at risk of fracture located at meta-epiphyseal level and meta-epiphyseal lesions with poor response to non-surgical therapies are a good indication for resection surgery and replacement with prosthesis. Osteosynthesis is indicated in pathological fractures or injuries at risk of pathological diaphyseal fractures. In the literature, there are several attempts of standardization of the treatments of BM (Capanna and Campanacci, SIOT guidelines) in absence of univocal guidelines [193,194]. The feasibility study is the first step to determine the type of treatment to be performed. Patients who are not eligible for excisional surgery are evaluated by a multidisciplinary team formed by oncologists, radiotherapists, orthopedic surgeons, and interventional radiologists.

## 4. Indications in Primary Bone Tumors

Indications

Treatment of primary malignant bone tumors (osteosarcoma, Ewing’s sarcoma, and chondrosarcoma), which account for less than 1% of tumors diagnosed each year, is primarily surgical, with accompanying systemic therapy if necessary. An interdisciplinary approach may employ percutaneous ablation, as in the case of lesions located in anatomically unfavorable, inoperable, or recurrent sites. However, malignant tumors are too large at the time of patient presentation to allow effective local ablative control of the tumor. Therefore, it seems reasonable to use radiofrequency ablation to treat locally recurrent malignancy or when resection is not feasible. Percutaneous curative treatment of primary bone tumors is feasible for benign lesions such as OO, OB, and chondroblastoma. OO was the first tumor to be successfully treated with RFA at any site. RFA is curative in most patients, usually after a single treatment. If performed correctly, the frequency of complications is extremely low.

Available evidence

**Nakatsuka et al.** [179] enrolled 20 patients with unresectable malignant bone tumors. No adverse events occurred in 19 of the 20 patients (95%). VAS scores decreased by ≥2 points in 11 of 13 patients (84.6%). Tumor response (complete or partial response) was achieved in 16 of 18 patients (88.9%). The 1-year overall survival rate was 60.9%.

**Yu et al.** [180] performed a retrospective analysis of 27 patients who, from 2006 to 2010, had unresectable local recurrence of osteosarcoma. After HIFU treatment, the response rate was 51.8% and the local disease control rate was 85.2%. Local disease control rates at 1, 2, and 3 years were 59.2%, 40.7%, and 33.1%, respectively. The median time free from progression of local disease was 14 months, the median time free of progression was 13 months, and the median time to overall survival was 21 months.

**Anselmetti et al.** [181] analyzed 106 patients with multiple myeloma undergoing percutaneous vertebroplasty. The median pretreatment VAS score of 9 (range 4–10) decreased significantly (*p* < 0.001) to 1 (range 0–9) after vertebroplasty. Median pretreatment Oswestry Disability Index (ODI) values of 82% (range 36–89%) improved significantly to 7% (range 0–82%) (*p* < 0.001). Use of analgesic medications after treatment decreased significantly with a statistically significant difference (*p* < 0.001).

Using HIFU, **Chen et al**. treated 80 patients with bone tumors (osteosarcoma, chondrosarcoma, Ewing sarcoma, giant cell tumor) [21]. In combination with systemic chemotherapy, the survival of patients with stage 2b tumor disease was thereby significantly improved. In addition to individual case reports describing the treatment of aneurysmal bone cysts or giant cell tumors, when pain is in the forefront of symptoms, percutaneous ablation can additionally be employed in the palliation of primary bone tumors, analogously to the treatment of bone metastases [49]. Compared with traditional surgery for malignant bone tumors, HIFU has the advantage of being noninvasive and can be used to treat large malignant bone tumors that cannot be removed surgically. Limb shape and succession are not affected by HIFU, and treatment can be repeated on patients who have not fully responded or had recurrences. HIFU treatment provides pain relief and can be used in conjunction with chemotherapy. **Li et al.** [182] analyzed 25 patients with malignant bone tumors before and after HIFU treatment. After HIFU treatment, 21 (87.5%) patients were completely pain free, and 24 (100%) experienced significant relief. The response rate was 84.6%. The survival rates at 1, 2, 3, and 5 years were 100.0%, 84.6%, 69.2%, and38.5%, respectively.

Ablation with high-intensity focused ultrasound is also an option for palliative treatment. In the study by **Chen et al.** [21], the 5-year survival rate for patients with osteosarcoma stage III disease (Enneking staging system) was 15.8%, which is like the reported survival rate (15.7%) for a group of patients undergoing incomplete surgical excision (38). These results indicate that high-intensity focused ultrasound offers another choice for patients with advanced stages of bone neoplasia. However, patients with tumors larger than 10 cm, nerves involved by the tumors, or tumors in different locations, such as pelvic tumors, growing toward the pelvic cavity and partially covered by the bowel, should not be considered for curative treatment.

## 5. Conclusions

Skeletal metastases influence the quality of life of patients with metastatic diseases. In these patients, especially those with short survival expectancy, the indications for surgical treatment are limited, while immediate pain relief and improvement of the functional status are important, and complications of treatments are unwanted. Decision making regarding potential surgery for a metastatic disease necessarily requires availability of reliable information about patient survival and quality of life [195,196]. Currently, such novel mini-invasive surgical treatments as embolization, thermal ablation, and cementoplasty are valuable techniques in the management of patients with painful bone metastases both in terms of pain palliation and disease control. Many are the advantages of IR: immediate reduction of post-procedural pain, even in radio-resistant tumors, as a result of thermal destruction of fibers and pain receptors and reduction of the tumor volume (tumor debulking); low complication rate; low hospitalization costs; more comfort for the patient; possibility to repeat the procedure with the same technique without damaging the surrounding structures; possibility of repeated and multiple treatments, also in combination with other IR techniques such as cementoplasty or RT [8].

IR can also significantly help the management patients with primary bone tumors, even if the literature is still limited in this field. All these techniques, along with systemic treatment, surgery, and RT, offer a variety of treatment options for musculoskeletal oncologic patients.

## Figures and Tables

**Figure 1 jcm-11-03265-f001:**
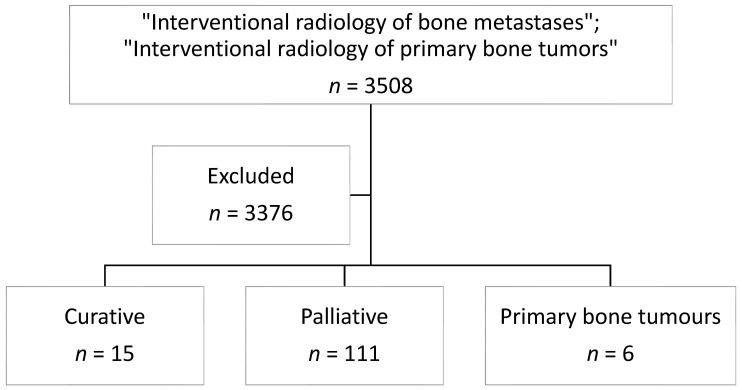
Flowchart summarizing the selection of studies in the literature.

**Table 1 jcm-11-03265-t001:** Techniques of IR and their applications.

Interventional Radiology Techniques				
Techniques	Type	Aim of the Technique	Indications	Curative Treatment
**Chemical ablation**	Ethanol, Doxycycline, Polidocanol (detergent sclerosant, STS (detergent sclerosant)	Chemical-based ablation	Malignant tumors	
**Embolization**	Nano-particles	Minimizing blood loss	To reduce blood deficit during surgery To lower pain and natural bleeding of tumors To reduce tumor vascularization before percutaneous ablation	
**RFA**	Energy-based ablations (thermoablation)	Ablation techniques (inducing tumoral necrosis) needle driven	Benign tumors Malignant tumors Neurovascular disease related to the musculoskeletal system	√
**MWA**	Energy-based ablations (thermoablation)	Ablation techniques (inducing tumoral necrosis) needle driven	Benign oo Malignant tumors	√
**CA**	Energy-based ablations (thermoablation)	Ablation techniques (inducing tumoral necrosis) needle driven	Benign Malignant neurovascular	√
**MRgFUS**	Energy-based ablations (thermoablation)	Ablation techniques (inducing tumoral necrosis) needleless technique	Benign oo malignant	√
**Cementoplasty**	Cementoplasty (PMMA) including free-hand injection, vertebroplasty, kyphoplasty	Mechanical stabilization techniques	Severe pain and neurological damage Spinal instability Pz with controindication to open surgery	
**Osteosynthesis**	Intramedullary nails and screws	Mechanical stabilization techniques	Pain palliation and stabilization of pathological bone segment	

**Table 2 jcm-11-03265-t002:** Results of curative ablation.

Author	Journal	Year	Ablation Modality	Title	No. of Patients/(no. of Tumors)	% Local Control	Survival Rate Overall Survival (%/Years; Median Month	Follow-Up (Months)	% Complications
**Cazzato R. et al.** [77]	Diagn Interv Imaging	2021	RFA, CA	Percutaneous thermal ablation of sacral metastases: Assessment of pain relief and local tumor control	23	7 (30%)	NR	21	22
**Cazzato R. et al.** [28]	International Journal of Hyperthermia	2018	RFA + CA	Percutaneous image-guided ablation of bone metastases: local tumor control in oligometastatic patients	46	71.7% 1 year	95.4% 2 years	34	10
**Ma Y. et al.** [78]	Cardiovasc Intervent Radiol	2018	CA, RFA, MWA	Percutaneous Image-Guided Ablation in the Treatment of Osseous Metastases from Non-small Cell Lung Cancer	45 (76)	68% 1 year	NR	NR	2.6
**Vaswani D. et al.** [79]	Cardiovasc Intervent Radiol	2018	RFA, CA	Radiographic Local Tumor Control and Pain Palliation of Sarcoma Metastases within the Musculoskeletal System with Percutaneous Thermal Ablation	13 (13)	100% 1 year	NR	12	5
**Gardner C. et al.** [15]	J Bone Joint Surg Am	2017	CA	Cryoablation of Bone Metastases from Renal Cell Carcinoma for Local Tumor Control	40 (50)	41/50 (82%)	31 (77/1 year 26/5 years)	35	4 (8)
**Erie A. et al.** [80]	J Vasc Interv Radiol	2017	RFA, CA	Retrospective Review of Percutaneous Image-Guided Ablation of Oligometastatic Prostate Cancer: A Single-Institution Experience	16 (18)	15 (83%)	100/2 years	27	0
**Aubry S. et al.** [81]	Eur Radiol	2017	MWA	Prospective 1-year follow-up pilot study of CT-guided microwave ablation in the treatment of bone and soft-tissue malignant tumours	13 (16)	4 (36.3%)	NR	12	0
**Wallace A. et al.** [13]	AJNR Am J Neuroradiol	2016	CA	Radiographic Local Control of Spinal Metastases with Percutaneous Radiofrequency Ablation and Vertebral Augmentation	56 (92)	79% 1 year	NR	NR	4.3
**Tomasian A. et al.** [82]	AJNR Am J Neuroradiol	2016	CA	Spine Cryoablation: Pain Palliation and Local Tumor Control for Vertebral Metastases	14 (31)	30 (96.7%)	NR	10	0
**Wallace A. et al.** [13]	AJNR Am J Neuroradiol	2016	RFA	Radiographic Local Control of Spinal Metastases with Percutaneous Radiofrequency Ablation and Vertebral Augmentation	NR (55)	70% 1 year	NR	7.9	0
**Barral M. et al.** [30]	Cardiovasc Intervent Radiol	2016	RFA, CRIO, MWA	Percutaneous Thermal Ablation of Breast Cancer Metastases in Oligometastatic Patients	79/114;18/NR	83/1 year; 76/2 years	98.3/1 year, 95.5/2 years	18.4	12 (10.5)
**Deschamps F. et al.** [83]	Eur Radiol	2014	RFA, CA	Thermal ablation techniques: a curative treatment of bone metastases in selected patients?	89 (122)	67% 1 year	91/1 year	22.8	11 (9)
**McMenomy B. et al** [84].	J Vasc Interv Radiol	2013	CA	Percutaneous cryoablation of musculoskeletal oligometastatic disease for complete remission	40 (52)	45 (87%)	91/1 year, 84/2 years	21	2 (5)
**Littrup P. et al.** [85].	J Vasc Interv Radiol	2013	CA	Soft-tissue cryoablation in diffuse locations: feasibility and intermediate term outcomes	126/251	225 (90%)	NR	11	5 (2.3)
**Bang H. et al.** [48].	J Vasc Interv Radiol	2012	CA	Percutaneous cryoablation of metastatic lesions from non-small-cell lung carcinoma: initial survival, local control, and cost observations	8 (18)	17 (94%)	NR	11	2 (11)
**Bang H. et al.** [86].	J Vasc Interv Radiol	2012	CA	Percutaneous cryoablation of metastatic renal cell carcinoma for local tumor control: feasibility, outcomes, and estimated cost-effectiveness for palliation	27 (48)	47 (97%)	NR	16	1 (2)

NR not reported, CA cryoablation, RFA radiofrequency ablation, MWA Microwave ablation.

**Table 3 jcm-11-03265-t003:** Results of palliative ablation and combined of the treatment of bone fracture and impending fracture, ***main results***.

Author	Journal	Year	Ablation Modality	Title	No. of Patients (No. of Tumors)	No. (%) of Patients with Reduced Pain	Mean Pain Score Change	Survival Rate Overall Survival (%/Years; Median Month)	Follow-Up (Months)	% Major Complications
**Arrigoni F. et al.** [87]	Radiol Med	2022	CA	CT-guided cryoablation for management of bone metastases: a single center experience and review of the literature	28	100%	6.9–3.5 (3.4/10)	100% 3 months	3	0
**Wang F. et al.** [16]	Skeletal Radiol	2022	RFA + PVP	The combination of radiofrequency ablation and vertebroplasty shows advantages over single vertebroplasty in treating vertebral neoplastic lesions	35	NR	8.46–1.7 (6.76/10)	NR	6	0
**Jennings J. et al.** [88]	Radiol Imaging Cancer	2021	CA	Cryoablation for Palliation of Painful Bone Metastases: The MOTION Multicenter Study	66	74%	7.3–3.7 (3.6/10)	NR	6	4.6
**Madani K. et al.** [89]	Support Care Cancer	2021	RFA + PVP	Combined local treatments for vertebral metastases with limited epidural extension	18 (24)	66.7%	7.3–2 (5.3/10)	73% 2 years	16.7	0
**Kastler A. et al.** [90]	Medicina (Kaunas)	2021	RFA + PVP	Bipolar Radiofrequency Ablation of Painful Spinal Bone Metastases Performed under Local Anesthesia: Feasibility Regarding Patient’s Experience and Pain Outcome	25	83%	8.4–1.8 (6.6/10)	100% 1 year	12	0
**Pusceddu C. et al.** [91]	Curr Oncol	2021	PVP + RFA	The Role of a Navigational Radiofrequency Ablation Device and Concurrent Vertebral Augmentation for Treatment of Difficult-to-Reach Spinal Metastases	35 (41)	NR	5.7–0.9 (4.8/10)	72% 1 year	19	0
**Levy J. et al.** [14]	J Vasc Interv Radiol	2020	RFA	Radiofrequency Ablation for the Palliative Treatment of Bone Metastases: Outcomes from the Multicenter OsteoCool Tumor Ablation Post-Market Study (OPuS One Study) in 100 Patients	100 (134)	100%	8.2–3.5 (4.7/10)	70% 6 months	6	4
**Jiao D. et al.** [18]	Acad Radiol	2020	MWA + PC	Simultaneous C-arm Computed Tomography-Guided Microwave Ablation and Cementoplasty in Patients with Painful Osteolytic Bone Metastases: A Single-center Experience	30 (42)	100%	7.4–1.3 (6.1/10)	66.7 1 year	12	0
**Yang Y. et al.** [92]	Cryobiology	2020	CA	Retrospective analysis of CT-guided percutaneous cryoablation for treatment of painful osteolytic bone metastasis	26 (36)	94.4%	7.1–1.8 (5.3/10)	100% 6 months	6	11,5
**Yang X. et al.** [93]	Eur J Radiol	2020	PVP	Vesselplasty using the Mesh-Hold™ bone-filling container for the treatment of pathological vertebral fractures due to osteolytic metastases: A retrospective study	63 (105)	97.4%	8.2–2.1 (6.1/10)	66.7% 1 year	4–30 months	16.2 cement leakage rate;1.9 paravertebral vein embolism
**De Marini P.et al.** [94]	Int J Hyperthermia	2020	RFA + CA	Percutaneous image-guided thermal ablation of bone metastases: a retrospective propensity study comparing the safety profile of radio-frequency ablation and cryo-ablation	274	NR	NR	99,64% 1 year	18.5	2.5
**Deib G. et al.** [95]	AJR Am J Roentgenol	2019	MWA + PVP	Percutaneous Microwave Ablation and Cementoplasty: Clinical Utility in the Treatment of Painful Extraspinal Osseous Metastatic Disease and Myeloma	65 (77)	NR	6.32–2 (4,32/10)	90.7% 1 year	6	0
**Tanigawa N. et al.** [96]	Cardiovasc Intervent Radiol	2018	RFA	Phase I/II Study of Radiofrequency Ablation for Painful Bone Metastases: Japan Interventional Radiology in Oncology Study Group 0208	33 (33)	69.7%	6–1.2 (4.8/10)	97% 1 year	12	12
**Prologo J. et al.** [97]	Skeletal Radiol	2014	RFA	Image-guided cryoablation for the treatment of painful musculoskeletal metastatic disease: a single-center experience	50 (54)	94%	8–3 (5/10)	100% 3 months	3	8
**Pusceddu C. et al.** [76]	J Vasc Interv Radiol	2013	MWA	Treatment of bone metastases with microwave thermal ablation	18 (21)	100%	5.6–0.45 (5.15/10)	100% 3 months	3	0
**Rossi G. et al.** [98]	Radiol Med	2013	TAE using N-2-butyl cyanoacrylate (NBCA)	Embolisation of bone metastases from renal cancer	107 (163)	96%	NR	10 months	48	Post embolisation syndrome 9.2%; Transient paraesthesias 25%; 1 intraprocedural tear of the left L3 artery and iliopsoas haemorrhage

NR not reported, CA cryoablation, RFA radiofrequency ablation, MWA Microwave ablation, HIFU High-intensity focused ultrasound, TAE Transarterial embolization, PVP Percutaneous vertebroplasty, PC percutaneous cementoplasty.

**Table 4 jcm-11-03265-t004:** Results of palliative ablation and combined of the treatment of bone fracture and impending fracture.

Author	Journal	Year	Ablation Modality	Title	No. of Patients(no. of Tumors)
**Cazzato R. et al.** [77]	Diagn Interv Imaging	2021	RFA/CA	Percutaneous thermal ablation of sacral metastases: Assessment of pain relief and local tumor control follow-up	23
**Pusceddu C. et al.** [17]	Medicina (Kaunas)	2021	MWA + PC	Combined Microwave Ablation and Osteosynthesis for Long Bone Metastases	11
**Campanacci L. et al.** [99]	Eur J Surg Oncol	2021	ECT	Operating procedures for electrochemotherapy in bone metastases: Results from a multicenter prospective study on 102 patients	102
**Koirala N. et al.** [100]	Skeletal Radiol	2020	PC	Percutaneous reinforced osteoplasty for long bone metastases: a feasibility study	15
**Giles S. et al.** [101]	J Vasc Interv Radiol	2019	HIFU	Comparison of Imaging Changes and Pain Responses in Patients with Intra- or Extraosseous Bone Metastases Treated Palliatively with Magnetic Resonance-Guided High-Intensity-Focused Ultrasound	21
**Sundararajan S. et al.** [102]	J Oncol	2019	TAE+CRIO	Sequential Interventional Management of Osseous Neoplasms via Embolization, Cryoablation, and Osteoplasty cooling system	15
**Tian Q. et al.** [103]	Korean J Radiol	2019	PSP	Percutaneous Sacroplasty for Painful Sacral Metastases Involving Multiple Sacral Vertebral Bodies: Initial Experience with an Interpedicular Approach	10
**Liu H. et al.** [104]	Cardiovasc Intervent Radiol	2019	POP	Application of Percutaneous Osteoplasty in Treating Pelvic Bone Metastases: Efficacy and Safety	126
**Cazzato R. et al.** [105]	Int J Hyperthermia	2018	RFA	Low-power bipolar radiofrequency ablation and vertebral augmentation for the palliative treatment of spinal malignancies	11 (11)
**Chen Z. et al.** [106]	Orthop Surg	2018	HIFU	Evaluation of Quality of Life Using EORTC QLQ-BM22 in Patients with Bone Metastases after Treatment with Magnetic Resonance Guided Focused Ultrasound	26
**Khan M. et al.** [107]	AJNR	2018	MWA	Efficacy and safety of percutaneous microwave ablation and cementoplasty in the treatment of painful spinal metastases and myeloma	69 (102)
**Couraud G. et al.** [108]	J bone oncol	2018	PC	Evaluation of short-term efficacy of extraspinal cementoplasty for bone metastasis: a monocenter study of 31 patients.	31
**Fares A. et al.** [109]	J Egypt Natl Canc Inst	2018	PC	Combined percutaneous radiofrequency ablation and cementoplasty for the treatment of extraspinal painful bone metastases: a prospective study. J Egypt Natl Canc	30
**Bertrand A. et al.** [110]	J Ther Ultrasound	2018	HIFU	Focused ultrasound for the treatment of bone metastases: effectiveness and feasibility	17
**Ma Y. et al.** [78]	Cardiovasc Intervent Radiol	2018	RFA/CA/MWA	Percutaneous Image-Guided Ablation in the Treatment of Osseous Metastases from Non-small Cell Lung Cancer	45
**Vaswani D. et al.** [79]	Cardiovasc Intervent Radiol	2018	CA + PC	Radiographic Local Tumor Control and Pain Palliation of Sarcoma Metastases within the Musculoskeletal System with Percutaneous Thermal Ablation	41
**Coupal T. et al.** [111]	Pain Physician	2017	TAE	The Hopeless Case? Palliative Cryoablation and Cementoplasty Procedures for Palliation of Large Pelvic Bone Metastases	48
**Pusceddu C. et al.** [112]	Skeletal Radiol	2017	PC	CT-guided percutaneous screw fixation plus cementoplasty in the treatment of painful bone metastases with fractures or a high risk of pathological fracture	27
**Reyad R. et al.** [113]	Diagn Interv Imaging	2017	PVP	Thick cement usage in percutaneous vertebroplasty for malignant vertebral fractures at high risk for cement leakage	77
**Motta A. et al.** [114]	Eur J Radiol	2017	CA	Feasibility of percutaneous cryoablation of vertebral metastases under local anaesthesia in ASAIII patients	11
**McArthur T. et al.** [115]	Curr Probl Diagn Radiol	2017	CA	Percutane Image-Guided Cryoablation of Painful Osseous Metastases: A Retrospective Single-Center Review	16
**Baagla S. et al.** [116]	Cardiovasc Intervent Radiol	2016	RFA	Multicenter Prospective Clinical Series Evaluating Radiofrequency Ablation in the Treatment of Painful Spine Metastases	50 (69)
**Tomasian A. et al.** [82]	AJNR Am J Neuroradiol	2016	CA	Spine Cryoablation: Pain Palliation and Local Tumor Control for Vertebral Metastases	14 (31)
**Facchini G. et al.** [117]	Eur J Orthop Surg Traumatol	2016	TAE	Palliative embolization for metastases of the spine	164
**Pusceddu C. et al.** [118]	Cardiovasc Intervent Radiol	2016	MWA + PC	Combined Microwave Ablation and Cementoplasty in Patients with Painful Bone Metastases at High Risk of Fracture follow-up	35 (37)
**Anzidei M. et al.** [119]	Radiol Med	2016	HIFU	Magnetic resonance-guided focused ultrasound for the treatment of painful bone metastases: role of apparent diffusion coefficient (ADC) and dynamic contrast enhanced (DCE) MRI	23
**Wang F. et al.** [120]	Pain Physician	2016	TAE	Sequential Transarterial Embolization Followed by Percutaneous Vertebroplasty Is Safe and Effective in Pain Management in Vertebral Metastases	25
**Chen F. et al.** [121]	Oncol Lett	2016	RFA	Percutaneous kyphoplasty for the treatment of spinal metastases	282
**Susa M. et al.** [47]	BMC Cancer	2016	CA	CT guided cryoablation for locally recurrent or metastatic bone and soft tissue tumor: initial experience	9
**Bianchi G. et al.** [122]	World J Surg	2016	ECT	Electrochemotherapy in the Treatment of Bone Metastases: A Phase II Trial	29
**Jiao D. et al.** [123]	Oncotarget	2016	RFA	Radiofrequency ablation versus 125I-seed brachytherapy for painful metastases involving the bone	79
**Joo B. et al.** [124]	Yonsei Med J	2015	HIFU	Pain palliation in patients with bone metastases using magnetic resonance-guided focused ultrasound with conformal bone system: a preliminary report	5
**Wallace A. et al.** [125]	J Neurooncol	2015	RFA	Radiofrequency ablation and vertebral augmentation for palliation of painful spinal metastases	72 (110)
**Wei Z. et al.** [126]	Skeletal Radiol	2015	MWA	Computed tomography-guided percutaneous microwave ablation combined with osteoplasty for palliative treatment of painful extraspinal bone metastases from lung cancer	26 (33)
**Di Staso M. et al.** [127]	PLoS One	2015	CA + RTA	Treatment of solitary painful osseous metastases with radiotherapy, cryoablation or combined therapy: Propensity matching analysis in 175 patients	175
**Cazzato R. et al.** [128]	Eur J Surg Oncol	2015	RFA/PC/CA	Over ten years of single-institution experience in percutaneous image-guided treatment of bone metastases from differentiated thyroid cancer	25 (49)
**Tian Q. et al.** [129]	J Vasc Interv Radiol	2014	RFA + pop	Combination radiofrequency ablation and percutaneous osteoplasty for palliative treatment of painful extraspinal bone metastasis: a single-center experience	38
**Kastler A. et al.** [130]	J Vasc Interv Radiol	2014	MWA	Microwave thermal ablation of spinal metastatic bone tumors	17 (20)
**Hurwitz M. et al.** [131]	J Natl Cancer Inst	2014	HIFU	Magnetic resonance-guided focused ultrasound for patients with painful bone metastases: phase III trial results	112/NR versus 35/NR
**Alemann G. et al.** [132]	J Palliat Med	2014	RFA	Treatment of painful extraspinal bone metastases with percutaneous bipolar radiofrequency under local anesthesia: feasibility and efficacy in twenty-eight cases	28
**Botsa E. et al.** [133]	Ann Palliat Med	2014	RFA/MWA	CT image guided thermal ablation techniques for palliation of painful bone metastases	45
**Li Z. et al.** [134]	Chin Med J (Engl)	2014	PVP	Kyphoplasty versus vertebroplasty for the treatment of malignant vertebral compression fractures caused by metastases: a retrospective study	80
**Deschamps F. et al.** [83]	Eur Radiol	2014	RFA + CA	Thermal ablation techniques: a curative treatment of bone metastases in selected patients?	89
**Li F. et al.** [135]	Pathol Oncol Res.	2014	CA	An Effective Therapy to Painful Bone Metastases: Cryoablation Combined with Zoledronic Acid	56
**Sun G. et al.** [136]	Eur Radiol	2014	PC	Cementoplasty for man- aging painful bone metastases outside the spine	51
**Callstrom M. et al.** [137]	Cancer	2013	CA	Percutaneous image-guided cryoablation of painful metastases involving bone: multicenter trial	61 (69)
**Napoli A. et al.** [138]	invest Radiol	2013	HIFU	Primary pain palliation and local tumor control in bone metastases treated with magnetic resonance-guided focused ultrasound	18 (18)
**Anselmetti G. et al.** [139]	Pain Physician	2013	PVP	Percutaneous vertebral augmentation assisted by PEEK implant in painful osteolytic vertebral metastasis involving the vertebral wall: experience on 40 patients	40
**Trumm C. et al.** [140]	Skeletal Radiol	2013	PVP	CT fluoroscopy-guided vertebral augmentation with a radiofrequency-induced, high-viscosity bone cement (StabiliT(^®^)): technical results and polymethylmethacrylate leakages in 25 patients	25
**Kastler A. et al.** [141]	Pain Med	2013	MWA	Analgesic effects of microwave ablation of bone and soft tissue tumors under local anesthesia	15 (25)
**Masala S. et al.** [60]	Neuroradiology.	2012	CA	Combined use of percutaneous cryoablation and vertebroplasty with 3D rotational angiograph in treatment of single vertebral metastasis: Comparison with vertebroplasty.	23
**Iannessi A. et al.** [142]	Diagn Interv Imaging	2012	PC	Percutaneous cementoplasty for the treatment of extraspinal painful bone lesion: a prospective study.	20
**Rossi G. et al.** [143]	Radiol Med	2011	TAE	Selective arterial embolisation for bone tumours: experience of 454 cases	365 (454)
**Masala S. et al.** [144]	Singapore Med J	2011	RFA + VP/CA + VP	Percutaneous ablative treatment of metastatic bone tumours: visual analogue scale scores in a short-term series	30
**Thacker P. et al.** [46]	AJR Am J Roentgenol.	2011	CA	Palliation of painful metastatic disease involving bone with imaging-guided treatment: Comparison of patients’ immediate response to radiofrequency ablation and cryoablation	36
**Masala S. et al.** [145]	*Skeletal Radiol*.	2011	CA	Metabolic and clinical assessment of efficacy of cryoablation therapy on skeletal masses by 18F-FDG positron emission tomography/computed tomography (PET/CT) and visual analogue scale (VAS): Initial experience	20
**Masala S. et al.** [146]	Support Care Cancer	2011	PC	Percutaneus osteoplasty in the treatment of extraspinal painful multiple myeloma lesions	39
**Rossi G. et al.** [5]	J Vasc Interv Radiol	2011	TAE	Selective embolization with N-butyl cyanoacrylate for metastatic bone disease	243 (309)
**Dupuy D. et al.** [25]	Cancer	2010	RFA	Percutaneous radiofrequency ablation of painful osseous metastases: a multicenter American College of Radiology Imaging Network trial follow-up	55 (55)
**Kashima M. et al.** [147]	AJR Am J Roentgenol	2010	RFA	Radiofrequency ablation for the treatment of bone metastases from hepatocellular carcinoma	40
**Liberman B.et al.** [54]	Ann Surg Oncol	2009	HIFU	Pain palliation in patients with bone metastases using MR-guided focused ultrasound surgery: a multicenter study	31 (32)
**Carrafiello G. et al.** [148]	Radiol Med	2009	RFA	Percutaneous imaging-guided ablation therapies in the treatment of symptomatic bone metastases: preliminary experience	10
**Delpla A. et al.** [149]	Cardiovasc Intervent Radiol	2009	PVP	Preventive Vertebroplasty for Long-Term Consolidation of Vertebral Metastases	100
**Thanos L. et al.** [150]	Skeletal Radiol	2008	RFA	Radiofrequency ablation of osseous metastases for the palliation of pain	30
**Gianfelice D. et al.** [151]	Radiology	2008	HIFU	Palliative treatment of painful bone metastases with MR imaging--guided focused ultrasound	11-dic
**Basile A. et al.** [152]	Radiol Med	2008	RFA + PC	Cementoplasty in the management of painful extraspinal bone metastases: our experience	13
**Anselmetti G. et al.** [153]	Cardiovasc Intervent Radiol	2008	PC	Treatment of extraspinal painful bone metastases with percutaneous cementoplasty: a prospective study of 50 patients MDCT features, and treatment with RFA	50
**Trumm C. et al.** [154]	J Vasc Interv Radiol	2008	PVP	CT fluoroscopy-guided percutaneous vertebroplasty for the treatment of osteolytic breast cancer metastases: results in 62 sessions with 86 vertebrae treated	53
**Tuncali K. et al.** [155]	AJR Am J Roentgenol	2007	CA + HIFU	MRI-guided percutaneous cryotherapy for soft-tissue and bone metastases: initial experience	22
**Catane R. et al.** [55]	Ann Oncol	2007	HIFU	MR-guided focused ultrasound surgery (MRgFUS) for the palliation of pain in patients with bone metastases--preliminary clinical experience	113
**Forauer A. et al.** [156]	Acta Oncol	2007	TAE	Selective palliative transcatheter embolization of bony metastases from renal cell carcinoma	21
**Callstrom M. et al.** [157]	Radiology	2006	CA	Painful metastases involving bone: percutaneous image-guided cryoablation--prospective trial interim analysis	14
**Barragán-Campos H. et al.** [69]	Radiology	2006	PVP	Percutaneous vertebroplasty for spinal metastases: complications	117
**Weber C. et al.** [158]	Rofo	2006	PVP	[CT-guided vertebroplasty and kyphoplasty: comparing technical success rate and complications in 101 cases]	69 (101)
**Callstrom M. et al.** [159]	Oncology	2005	RFA	Percutaneous ablation: safe, effective treatment of bone tumors	14
**Cai H. et al.** [160]	Ai Zheng	2005	RFA + VP/CA + VP	[Treatment effect of percutaneous vertebroplasty combined with interventional chemotherapy on vertebral metastases]	75
**Masala S. et al.** [161]	In Vivo	2005	PVP	MRI and bone scan imaging in the preoperative evaluation of painful vertebral fractures treated with vertebroplasty and kyphoplasty	30
**Guzman R. et al.** [162]	Eur Spine J	2005	TAE	Preoperative transarterial embolization of vertebral metastases	24
**Goetz M. et al.** [37]	J Clin Oncol	2004	RFA	Percutaneous image-guided radiofrequency ablation of painful metastases involving bone: a multicenter study	43 (43)
**Masala S. et al.** [163]	J Chemother	2004	PVP	Vertebroplasty and kyphoplasty in the treatment of malignant vertebral fractures	33
**Poggi G. et al.** [164]	Anticancer Res	2003	RFA	Percutaneous ultrasound-guided radiofrequency thermal ablation of malignant osteolyses	5
**Eustatia-Rutten C. et al.** [165]	J Clin Endocrinol Metab	2003	TAE	Outcome of palliative embolization of bone metastases in diferentiated thyroid carcinoma	16
**Fourney D. et al.** [166]	J Neurosurg	2003	PVP	Percutaneous vertebroplasty and kyphoplasty for painful vertebral body fractures in cancer patients	56
**Winking M. et al.** [167]	Ger Med Sci	2003	PVP	PMMA vertebroplasty in patients with malignant vertebral destruction of the thoracic and lumbar spine	22
**Alvarez L. et al.** [168]	Eur Spine J	2003	PVP	Vertebroplasty in the treatment of vertebral tumors: postprocedural outcome and quality of life	21
**Hierholzer J. et al.** [169]	J Vasc Interv Radiol	2003	PC	Percutaneous osteoplasty as a treatment for painful malignant bone lesions of the pelvis and femur	5
**Grönemeyer D. et al.** [170]	Cancer J	2002	RFA	Image-guided radiofrequency ablation of spinal tumors: preliminary experience with an expandable array electrode	10
**Chatziioannou A. et al.** [171]	Eur Radiol	2000	TAE	Preoperative embolization of bone metastases from renal cell carcinoma	26
**Barr J. et al.** [172]	Spine (Phila Pa 1976)	2000	PVP	Percutaneous vertebroplasty for pain relief and spinal stabilization	47
**Martin J. et al.** [173]	Bone	1999	PVP	Vertebroplasty: clinical experience and follow-up results	40
**Sun S. et al.** [174]	J Vasc Interv Radiol	1998	TAE	Bone metastases from renal cell carcinoma: preoperative embolization	16
**Weill A. et al.** [175]	Radiology	1996	PVP	Spinal metastases: indications for and results of percutaneous injection of acrylic surgical cement	37
**Cotten A. et al.** [176]	Radiology	1996	PVP	Percutaneous vertebroplasty for osteolytic metastases and myeloma: effects of the percentage of lesion filling and the leakage of methyl methacrylate at clinical follow-up	37
**Breslau J. et al.** [177]	J Vasc Interv Radiol	1995	TAE	Preoperative embolization of spinal tumors	14
**Corcos G. et al.** [178]	Spine (Phila Pa 1976)	1995	PVP	Cement leakage in percutaneous vertebroplasty for spinal metastases: a retrospective evaluation of incidence and risk factors	56

CA cryoablation, RFA radiofrequency ablation, MWA Microwave ablation, HIFU High-intensity focused ultrasound, TAE Transarterial embolization, PVP Percutaneous vertebroplasty, PC percutaneous cementoplasty, ECT eletrochemotherapy.

**Table 5 jcm-11-03265-t005:** Results of primary bone tumors.

Author	Journal	Year	Ablation Modality	Title	No. of Patients/(No. of Tumors)	% Local Control Tumor Response (Complete or Partial Response)	Mean Pain Score Change	No. (%) of Patients with Reduced Pain	Survival Rate Overall Survival (%/Years; Median Month	Follow-Up (Months)	% Major Complications
**Nakatsuka A. et al.** [179]	J Vasc Interv Radiol	2016	RFA	Safety and Clinical Outcomes of Percutaneous Radiofrequency Ablation for Intermediate and Large Bone Tumors Using a Multiple-Electrode Switching System: A Phase II Clinical Study	20	88.9%	3.6–1.5 (2.1/10)	84.6%	60.9% 1 year; 14.1 months	12	0
**Yu W. et al.** [180]	Surg Oncol	2015	HIFU	High-intensity focused ultrasound: noninvasive treatment for local unresectable recurrence of osteosarcoma	27	85.2%	2.04–0.32 (1.7/3)	NR	21 months	21	0
**Anselmetti G. et al.** [181]	Cardiovasc Intervent Radiol	2012	PVP	Percutaneous vertebroplasty in multiple myeloma: prospective long-term follow-up in 106 consecutive patients	106	NR	9–1 (8/10)	90%	NR	28.2 ± 12.1 months	1.6%
**Li C. et al.** [182]	Cancer	2010	HIFU	Noninvasive treatment of malignant bone tumors using high-intensity focused ultrasound	13	46.2%complete response; 38.4% partial response	1.85–0.08 (1.77/3)	100%	38.5% 5 year; 43.0 months	60	0
**Chen W. et al.** [21]	Radiology	2010	HIFU	Primary bone malignancy: effective treatment with high-intensity focused ultrasound ablation follow-up	80	86%	NR	NR	50.5% 5 year	60	50%
**Li C. et al.** [75]	Cancer Biol Ther	2009	HIFU	Osteosarcoma: Limb salvaging treatment by ultrasonographically guided high-intensity focused ultrasound	7	42.9% complete response;42.9% partial response	NR	100%	71.4% 5 year; 68 months	81	0

NR not reported, RFA radiofrequency ablation, HIFU High-intensity focused ultrasound, PVP percutaneous vertebroplasty.

## Data Availability

Not applicable.

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
