# Peer review of "Interventional Radiology in the Management of Metastases and Bone Tumors"

_jcm, 2022, doi:10.3390/jcm11123265_

Round 1

Reviewer 1 Report

  1. The author reviewed the literatures of Interventional radiology in the management of metastases and bone tumors. Overall, this review provided useful information for the reader to understand this technique.
  2. Abstract: Please describe it more clearly on the techniques of IR and its application, eg. Embolization, Thermal ablation, Radiofrequency Ablation……
  3. Please provide a table including different techniques of IR and their applications (indications).
  4. In the section of “Literature search strategy and results”, the author described the studies too much detail, eg. Arrigoni et al [78]…, Cazzato et al [79]…, Ma et al [80]…, Cazzato et al [82]…, Gardner et al [83]….. The author should summarize your key points, and put the citation accordingly, but not list the results of all studies. If possible, please do statistical analysis based on the data of previous studies.
  5. We advise the author divide the paper into different sections according to the different technique of IR. The results of literature search should be included in these sections.
  6. Table 1 and Table 2 just list the citations of the techniques and the number of patients, which provided very limited information. It is better to remove them.
  7. The author should provide some tables of figures of meta-analysis of the data extract from the literatures.

Author Response

# Reviewer 1 Comments for Author

1) The author reviewed the literatures of Interventional radiology in the management of metastases and bone tumors. Overall, this review provided useful information for the reader to understand this technique.

            >> Thank you

2) Abstract: Please describe it more clearly on the techniques of IR and its application, eg. Embolization, Thermal ablation, Radiofrequency Ablation……

            >> Done

3) Please provide a table including different techniques of IR and their applications (indications).

            >> Done

4) In the section of “Literature search strategy and results”, the author described the studies too much detail, eg. Arrigoni et al [78]…, Cazzato et al [79]…, Ma et al [80]…, Cazzato et al [82]…, Gardner et al [83]….. The author should summarize your key points, and put the citation accordingly, but not list the results of all studies. If possible, please do statistical analysis based on the data of previous studies.

            >> We proposed a more schematic approach because actually the study are very eterogenous and authors use multiple and different criteria to evaluate patients and results. In this way, it is not easy to to standardize the various results for a statistical analysis: in order to present the most recent and important literature data, we have decided to report them schematically.

5) We advise the author divide the paper into different sections according to the different technique of IR. The results of literature search should be included in these sections.

            >> yes, we observed this indication to divide results of curative vs palliative vs palliative with stabilization with cement because these are the three main purpose of the IR. It is not appropriate to divide the technique of ablation among cryoablation, RFA or MWA because it depends on the operator's experience and availability of materials

6) Table 1 and Table 2 just list the citations of the techniques and the number of patients, which provided very limited information. It is better to remove them.

            >> According also with other reviewers, we enriched these tables with clinical information.

7) The author should provide some tables of figures of meta-analysis of the data extract from the literatures.

            >> Because this is a narrative review, we didn’t perform a meta-analysis.

Reviewer 2 Report

The review Sgalambro et al. is devoted to the methods of interventional radiology, which are used in curative and palliative treatment of bone metastases and primary bone tumors. This review is presented in an unusual format and makes an ambiguous impression.

  1. In the Introduction section, it is necessary to explain the concept of “interventional radiology” in general and add its definition. The types of interventional radiology (diagnostic, therapeutic) and specific types of interventional radiology used in bone pathology should be listed in the Introduction section.
  2. Moreover, I would suggest creating a scheme that could clearly demonstrate the types of interventional radiology used in bone pathology and described in the first part of the Review.
  3. In my opinion, more attention should be paid to the first part of the Review, where interventional radiology techniques are described. Data from individual articles in the second part of the Review should not be given in such detail.
  4. Line 100 – an additional subheading that is not necessary. It will be clearer if the authors define this procedure in the first paragraph of this section (line 101).
  5. Abbreviations must be introduced at the first mention of the procedure name in the manuscript. For example, in line 118 the abbreviations are not spelled out, but the first mention of the procedure name appears in line 123; in line 147 the abbreviations are not deciphered, although the first mention of these names appeared earlier (line 144) without the introduction of abbreviations.
  6. Lines 111, 160, 166, 167, 185, 203, 208, 216, 234, 251, 264, 265, 270, and 293 – there are no spaces between words and reference numbers.
  7. Line 135 – misprint in the word “relapse”; line 554 – misprint in the reference number “al[99]al”.
  8. It is necessary to describe in more detail the methodology of the RFA and HIFU procedures (sections 2.1 and 2.4).
  9. All paragraphs concerning Indications should be uniform throughout the entire manuscript.
  10. Lines 181,182, 188 – US, MRI, and AVM are not deciphered.
  11. In the second part of the Review, the authors studied 134 articles. The same number of articles is shown in tables 1-3. However, the bibliography contains only 113 references.
  12. I recommend moving Figure 1 to Supplementary Materials as it's not really needed in the main manuscript. In addition, I did not find a figure legend.
  13. Lines 300, 318, 324, and 325 – abbreviations are being deciphered again, although the names of the procedures have already been introduced earlier. Please make decoding of the abbreviations at the first mention of any procedure name in the manuscript and then use only abbreviations.
  14. Line 367 – CI is mentioned earlier, but deciphered only now.
  15. Lines 371, 415, 428, 437, 467, 584, 604, 667, 672, 679, and 704 – make the authors of the mentioned articles in bold type.
  16. In 2.2 Section, reference style does not comply with the journal requirements. Please, correct it.
  17. Line 463 – what does (A) mean?
  18. Lines 493-494 – the level of the subheading is not clear, since it stands out from the general style of the Review.
  19. Line 650 – subheading style does not match the style of subheadings of the same level.
  20. Footnote to Table 2 – a consistent abbreviation style should be use.
  21. The formatting and style of the Table 3 is different from those in Tables 1 and 2.
  22. The reference list is not designed in a uniform style. For example, Ref1 – only the first author is specified, Ref2 – all authors are listed.

Author Response

# Reviewer 2 Comments for Author

The review Sgalambro et al. is devoted to the methods of interventional radiology, which are used in curative and palliative treatment of bone metastases and primary bone tumors. This review is presented in an unusual format and makes an ambiguous impression.

1) In the Introduction section, it is necessary to explain the concept of “interventional radiology” in general and add its definition. The types of interventional radiology (diagnostic, therapeutic) and specific types of interventional radiology used in bone pathology should be listed in the Introduction section.

            >> We added the definition of IR; the specific types of IR are discussed in the specific paragraph.

2) Moreover, I would suggest creating a scheme that could clearly demonstrate the types of interventional radiology used in bone pathology and described in the first part of the Review.

            >> According also with other reviewers, we add a table.

3) In my opinion, more attention should be paid to the first part of the Review, where interventional radiology techniques are described. Data from individual articles in the second part of the Review should not be given in such detail.

            >> Other reviewer suggested the opposite: however, we reviewed this part in order to offer a description of the techniques for useful for a general public.

4) Line 100 – an additional subheading that is not necessary. It will be clearer if the authors define this procedure in the first paragraph of this section (line 101).

            >> Done

5) Abbreviations must be introduced at the first mention of the procedure name in the manuscript. For example, in line 118 the abbreviations are not spelled out, but the first mention of the procedure name appears in line 123; in line 147 the abbreviations are not deciphered, although the first mention of these names appeared earlier (line 144) without the introduction of abbreviations.

            >> Done

6) Lines 111, 160, 166, 167, 185, 203, 208, 216, 234, 251, 264, 265, 270, and 293 – there are no spaces between words and reference numbers.

            >> Done

7) Line 135 – misprint in the word “relapse”; line 554 – misprint in the reference number “al[99]al”.

            >> Done

8) It is necessary to describe in more detail the methodology of the RFA and HIFU procedures (sections 2.1 and 2.4).

            >> Other reviewer suggested the opposite. We reviewed all this section in order to offer a description of the techniques for useful for a general public.

9) All paragraphs concerning Indications should be uniform throughout the entire manuscript.

            >> Done

10) Lines 181,182, 188 – US, MRI, and AVM are not deciphered.

            >> done

11) In the second part of the Review, the authors studied 134 articles. The same number of articles is shown in tables 1-3. However, the bibliography contains only 113 references.

            >> In the biblio, we cited only the paper discussed in the text.

12) I recommend moving Figure 1 to Supplementary Materials as it's not really needed in the main manuscript. In addition, I did not find a figure legend.

            >> thanks, we will consider this suggestion.

13) Lines 300, 318, 324, and 325 – abbreviations are being deciphered again, although the names of the procedures have already been introduced earlier. Please make decoding of the abbreviations at the first mention of any procedure name in the manuscript and then use only abbreviations.

            >> Done

14) Line 367 – CI is mentioned earlier, but deciphered only now.

15) Lines 371, 415, 428, 437, 467, 584, 604, 667, 672, 679, and 704 – make the authors of the mentioned articles in bold type.

16) In 2.2 Section, reference style does not comply with the journal requirements. Please, correct it.

            >> Done

17) Line 463 – what does (A) mean?

            >> modified

18) Lines 493-494 – the level of the subheading is not clear, since it stands out from the general style of the Review.

            >> Done

19) Line 650 – subheading style does not match the style of subheadings of the same level.

            >> Done

20) Footnote to Table 2 – a consistent abbreviation style should be use.

21) The formatting and style of the Table 3 is different from those in Tables 1 and 2.

22) The reference list is not designed in a uniform style. For example, Ref1 – only the first author is specified, Ref2 – all authors are listed.

            >> these points were revised.

Reviewer 3 Report

The review article entitle "Interventional radiology in the management of metastases and bone tumors" written by Sgalambro et al., has well described the interventional radiology for the primary malignancies of bone and they also illustrated the advantages and limitation of therapy, this review article will help the basic scientist and physician for apply their advancing the research and therapeutic aspects,

I strongly recommend for consideration, 

Thank you

Author Response

# Reviewer 3 Comments for Author

The review article entitle "Interventional radiology in the management of metastases and bone tumors" written by Sgalambro et al., has well described the interventional radiology for the primary malignancies of bone and they also illustrated the advantages and limitation of therapy, this review article will help the basic scientist and physician for apply their advancing the research and therapeutic aspects,

I strongly recommend for consideration

            >> Thank you for your comment.

Round 2

Reviewer 1 Report

Thanks for the authors. The revision is satisfactory.

Author Response

Thank you for your comment.

Reviewer 2 Report

Only some of the suggestions were taken into account, therefore there are comments to the manuscript that require corrections.

1.  All tables continue to be in a diverse format.

2. Figure caption is absent.

3. Indications sections continue to be styled differently throughout the manuscript.

4. The references continue to be in different formats.

Author Response

Dear Reviewer, thank you for your comments.

Here the replies: 

  1. All tables continue to be in a diverse format.
    >> We revised the style of the text and the format of the tables.

2. Figure caption is absent.
>> We add the caption.

3. Indications sections continue to be styled differently throughout the manuscript.
>> We revised this point.

4. The references continue to be in different formats.
>> the formats have been uniformed.